# Comparison of Simultaneous Single-Position Oblique Lumbar Interbody Fusion and Percutaneous Pedicle Screw Fixation with Posterior Lumbar Interbody Fusion Using O-arm Navigated Technique for Lumbar Degenerative Diseases

**DOI:** 10.3390/jcm10214938

**Published:** 2021-10-26

**Authors:** Ying Tan, Masato Tanaka, Sumeet Sonawane, Koji Uotani, Yoshiaki Oda, Yoshihiro Fujiwara, Shinya Arataki, Taro Yamauchi, Tomoyuki Takigawa, Yasuo Ito

**Affiliations:** 1Department of Orthopaedic Surgery, Okayama Rosai Hospital, Okayama 702-8055, Japan; tanying335@163.com (Y.T.); drsumeet166@gmail.com (S.S.); coji.uo@gmail.com (K.U.); odaaaaaaamn@yahoo.co.jp (Y.O.); fujiwarayoshihiro2004@yahoo.co.jp (Y.F.); araoyc@gmail.com (S.A.); ygitaro0307@yahoo.co.jp (T.Y.); 2Department of Spinal Surgery, Weifang Traditional Chinese Medicine Hospital, Weifang 261041, China; 3Department of Orthopaedic Surgery, Kobe Red Cross Hospital, Hyogo 651-0073, Japan; takigawa2004@yahoo.co.jp (T.T.); y-ito@kobe.jrc.or.jp (Y.I.)

**Keywords:** O-arm navigation, simultaneous single-position oblique lateral interbody fusion, indirect decompression

## Abstract

Minimally invasive posterior or transforaminal lumbar interbody fusion (MI-PLIF/TLIF) are widely accepted procedures for lumbar instability due to degenerative or traumatic diseases. Oblique lateral interbody fusion (OLIF) is currently receiving considerable attention because of the reductions in damage to the back muscles and neural tissue. The aim of this study was to compare clinical and radiographic outcomes of simultaneous single-position OLIF and percutaneous pedicle screw (PPS) fixation with MI-PLIF/TLIF. This retrospective comparative study included 98 patients, comprising 63 patients with single-position OLIF (Group SO) and 35 patients with MI-PLIF/TLIF (Group P/T). Cases with more than 1 year of follow-up were included in this study. Mean follow-up was 32.9 ± 7.0 months for Group SO and 33.7 ± 7.5 months for Group P/T. Clinical and radiological evaluations were performed. Comparing Group SO to Group P/T, surgical time and blood loss were 118 versus 172 min (*p* < 0.01) and 139 versus 374 mL (*p* < 0.01), respectively. Cage height, change in disk height, and postoperative foraminal height were significantly higher in Group SO than in Group P/T. The fusion rate was 96.8% in Group SO, similar to the 94.2% in Group P/T (*p* = 0.985). The complication rate was 6.3% in Group SO and 14.1% in Group P/T (*p* = 0.191). Simultaneous single position O-arm-navigated OLIF reduces the surgical time, blood loss, and time to ambulation after surgery. Good indirect decompression can be achieved with this method.

## 1. Introduction

Spinal instability due to degenerative, traumatic, infectious and neoplastic diseases may require fusion surgery [1]. Posterior lumbar interbody fusion (PLIF) and transforaminal lumbar interbody fusion (TLIF) have been widely accepted procedures for this purpose, but involve paraspinal muscles stripping, posterior vertebral bone removal, neural tissue handling and a risk of dural tear that over the long term may lead to chronic back pain and neurological defects [2]. To address these problems, minimally invasive PLIF/TLIF (MI-PLIF/TLIF) were introduced to reduce some of these complications [3]. Mayer described a less-invasive retroperitoneal pre-psoas approach, equivalent to oblique lumbar interbody fusion (OLIF) [4]. OLIF has advantages like indirect decompression with less exposure-related morbidity and less postoperative pain [5]. This technique also reduces blood loss, accelerates recovery, preserves posterior structures and requires no handling of neural tissues [6].

In conventional OLIF, cage insertion is performed with the patient in the lateral position, followed by percutaneous pedicle screw (PPS) insertion in the prone position. Repositioning of the patient on the operating table is required, which is indeed suboptimal and increases both operative time and medical costs [7]. To reduce the operative time, we started performing single-lateral position OLIF and posterior fixation with PPS [8]. To further reduce the operative time, we reported O-arm-navigated single-position OLIF with simultaneous PPS insertion performed by two surgeons in 2017 [9]. No studies appear to have compared clinical, surgical and radiographic outcomes of simultaneous single-position OLIF and PPS fixation with MI PLIF/TLIF performed under O-arm navigation. The aim of this study was to retrospectively compare clinical, surgical and radiographic outcomes for both techniques.

## 2. Materials and Methods

This study was approved by the ethics committee of our institute. We retrospectively evaluated patients who underwent lumbar interbody fusion between May 2017 and January 2020. Inclusion criteria were one-level fusion and more than one year of follow-up. Exclusion criteria were infection and current/history of spinal tumor. The 98 patients with lumbar degenerative disease who matched those criteria comprised 63 patients with simultaneous single-position OLIF-PPS (Group SO) and 35 patients with MI-PLIF/TLIF (7 patients with PLIF, 28 patients with TLIF: Group P/T). Group SO included 20 men and 43 women, while Group P/T included 15 men and 20 women.

### 2.1. Surgical Settings and Procedures for Single-Position OLIF

This procedure (Appendix A) is performed with the patient in the right lateral decubitus position with neuromonitoring. A hinged carbon fiber table is used to facilitate O-arm scan. The patient is kept at the posterior aspect of the table and secured with tape. The table is bent 15–20 degrees to open up the disc spaces. A bone graft is obtained from the posterior iliac crest and the reference frame is applied to the sacroiliac joint through the same incision. The O-arm scan is obtained and the images are transmitted to the Stealth station navigation system Spine7^R^ (Medtronic, Medtronic Sofamor Danek, Minneapolis, MN, USA). The navigation instruments are registered and a skin incision for the intended bilateral PPS and disc space is marked using a navigated pointer. Through a 2-cm incision, cranial PPSs are inserted on either side into the cranial vertebra (L4) with the help of navigated instruments by the first assisting surgeon. The main surgeon simultaneously proceeds with insertion of the OLIF cage. The navigated probe helps center the 4-cm left oblique skin incision. The fibers of the external oblique, internal oblique and transversus abdominis muscles are split during exposure. The navigated first dilator is placed anterior to psoas at the disc space level and sequential dilation is performed until a 22-mm tubular retractor can be placed. The self-retaining retractors are applied over the anterior aspect of the psoas muscle. Discectomy, cage insertion and PPS insertion are performed simultaneously by two separate surgeons using navigated instruments alternately (Figure 1 and Figure 2). The important point is that after placing the distal PPS, the cage is inserted because navigational accuracy will change if the OLIF cage is inserted first. The complete procedure has been described in a technical note published in 2019 [9].

### 2.2. Surgical Settings and Procedures for MI-PLIF/TLIF

In the MI-PLIF/TLIF group, with the use of navigated PLIF, patients undergo mini-open partial laminectomy, bilateral cage insertion with bone grafting, and posterior fixation with PPS. In navigated TLIF, patients undergo mini-open unilateral facetectomy, insertion of a bean cage with resected lamina tip bone graft insertion, and posterior fixation with PPS. All patients undergo fixation with navigated PPSs.

### 2.3. Clinical Evaluation

Clinical outcomes are assessed using values including visual analogue scale (VAS) for back pain and Oswestry Disability Index (ODI) [10]. This data was collected preoperatively and at 3, 6, 12 and 24 months postoperatively. The time to ambulation was documented in each group.

### 2.4. Surgical Evaluation

Surgical time, blood loss, time to ambulation postoperatively, any complications (including neurological deficit, dural tears, end plate fracture, infection, epidural hematoma, reoperation, implant failure and misplacement of implants) were noted.

### 2.5. Radiographic Evaluation

The following radiological outcomes were measured: pre- and postoperative disc height (DH); foraminal height (FH); foraminal area (FA); and segmental lordosis (SL). Change in DH was determined from computed tomography (CT). The cage height (CH) used during surgery was documented. DH was calculated as the mean value of anterior disc height (ADH) and posterior disc height (PDH), with ADH measured as the distance between the two endplates at the anterior aspect of the disc space and PDH measured as the distance between the two endplates at the posterior aspect of the disc. DH correction was calculated as CH minus the preoperative DH. FH was measured as the distance between the inferior pedicle of the cranial vertebra and the superior pedicle of the caudal vertebra (Figure 3). FA was measured using a digital tool measuring the area in the picture archiving and communication system (SYNAPSE5; Fujifilm Medical Systems, Lexington, MA, USA). All data were assessed by 2 senior spine surgeons (K.U. and Y.F.). Lumbar interbody union was evaluated in each group at the 1-year follow-up using CT.

### 2.6. Statistical Analysis

Continuous variables are presented as mean ± standard deviation (SD) and categorical variables are presented as counts. Comparison between groups was performed using the Mann-Whitney U test and chi-squared test, with values of *p* < 0.05 considered statistically significant. All analyses were performed using SPSS version 19.0 (IBM, Beijing, China).

## 3. Results

The demographic data and level of fusion for patients are shown in Table 1.

### 3.1. Clinical Evaluation

Pre- and postoperative clinical, surgical and radiographic data are summarized in Table 2 and Figure 4, Figure 5 and Figure 6. Mean follow-up period was 32.9 ± 7.0 months for Group SO and that 33.7 ± 7.5 months for Group P/T, ranging from 12 to 34 months in both groups. Changes in VAS and ODI did not differ significantly between groups.

### 3.2. Surgical Evaluation

Surgical time was lower in Group SO (112.0 ± 32.4 min) than in Group P/T (171.8 ± 40.6 min; *p* < 0.001). Blood loss was significantly lower in Group SO (139.2 ± 82.0 mL) than in Group P/T (374.2 ± 247.7 mL; *p* < 0.001) (Figure 3). The time to ambulation was significantly shorter in Group SO (2.7 ± 1.0 days) than in Group P/T (3.9 ± 2.4 days; *p* < 0.001).

In Group SO, one case showed infection at the site of reference frame incision that was treated with dressings and antibiotics, and two cases showed thigh pain and hip flexion weakness, all of which resolved within 2 months. Two cases in Group P/T showed dural tear and two cases had hematoma requiring evacuation surgery. The total complication rate was thus 6.3% in Group SO and 14.1% in Group P/T. These results are summarized in Table 3.

### 3.3. Radiographic Evaluation

In radiological parameters, preoperative differences in DH, FA, and SL were not significant between groups, while FH was significantly greater in Group SO (14.4 ± 4.5 mm) than in Group P/T (11.7 ± 5.3 mm; *p* = 0.021). CH used was significantly higher in Group SO (10.1 ± 1.3 mm) than in Group P/T (7. 8 ± 1.0 mm; *p* < 0.00001). DH correction was 5.2 ± 1.9 in Group SO and 2.5 ± 2.3 in Group P/T. CDR, postoperative DH, FH and FA were significantly higher in Group SO than in Group P/T, with percentage increases of 64% in DH, 25% in FH and 30% in FA in Group SO (Figure 5). The fusion rates of Group SO and Group P/T at the 1-year follow-up were 96.8% and 94.2%, respectively (*p* = 0.985) (Figure 6).

Representative follow-up radiograms for both groups are shown in Figure 7. In Group SO, a large cage was inserted and solid bony fusion was obtained.

## 4. Discussion

PLIF and TLIF are established procedures for lumbar fusion. However, these techniques are associated with complications such as paraspinal muscle injury, damage to posterior support structures, prolonged muscle retraction, difficulty in disc space visualisation and preparation, and the need for revision surgeries [1,11]. Such issues have led surgeons toward indirect decompression, which relies on restoration of DH. leading to increases in FH, and stretching of the ligamentum flavum and posterior longitudinal ligament to restore the central spinal canal [12]. OLIF and lateral lumbar interbody fusion (LLIF) provide advantages such as indirect neural decompression with solid bone fusion, the ability to insert a large cage, low risk of cage subsidence, and lower incidence of dural tears [1,7,12]. Compared to posterior procedures, restoration of the coronal and sagittal profiles is better [13,14]. One indirect decompression procedure is LLIF, which utilizes a trans-psoas route [15]. However, LLIF also shows unique disadvantages, such as lumbar plexus injury and psoas muscle weakness [16]. With conventional LLIF, C-arm usage is necessary and navigated implants and instruments are difficult to use. OLIF has received considerable attention, providing a clear operative field, reduced neurological complications, and the availability of navigated implants [1,14,17].

Traditionally, with the OLIF procedure, a cage is inserted in a lateral position and PPS is then performed in a prone position. This necessitates a change in position, leading to increased surgical time. Single-position OLIF has been proved to reduce surgery time by almost 60 min [7,8]. Another comparative study reported that navigated single-position OLIF reduced operative time by 30 min compared to repositioning OLIF [18]. With our technique, OLIF can be performed under navigation in a lateral position in a single sitting with simultaneous interbody fusion and PPS fixation. With our procedure, mean operative time was reduced by around 60 min compared to Group P/T. This reduced operative time reduces medical cost and allows faster turnover of cases. With OLIF, tissue trauma is reduced and posterior structures remain undamaged. For this reason, patients can be mobilized and discharged earlier [17]. The time for ambulation in our study was 2.7 ± 1.0 days for Group SO and 3.9 ± 2.4 days for Group P/T. The clinical results of ODI and VAS presented no difference between groups. This indicates that indirect decompression was effective for Group SO (OLIF cases). In our patients who underwent OLIF, DH increased by 64%, FH by 25% and FA by 30% due to indirect decompression. Lin et al. showed that after OLIF, DH increased by 49%, FH by 19%, and FA by 64% [6].

The total complication rate was 6.3% in Group SO and 14.1% in Group P/T. In the study by Lin et al., the complication rate was 36% in the OLIF group and 32% in TLIF group [17]. A meta-analysis reported that complication rate after OLIF was 26.7% with psoas weakness as the most common complication [19]. Fewer complications were seen in our OLIF cases compared to the literature [20,21,22]. This may be due to the use of neuromonitoring and O-arm navigation. Although the difference was not significant, Group P/T tended to show better correction of SL compared to Group SO in our patients. This may be because of the use of the lateral position for OLIF, which limits the potential for correction of lordosis.

Concern remains about radiation hazards for minimally invasive surgery (MIS) surgeons and staff working in high-volume centers. Our O-arm-navigated technique can be conducted without any radiation exposure to operating staff. The potential for increased radiation exposure to patients remains contentious, but an O-arm scan takes around 20–24 s, equivalent to 1.5 min of fluoroscopy [23]. One pedicle screw placement under fluoroscopy takes 7–20 s [24]. We further reduced radiation exposure to patients by setting a low field of view and low-dose mode for the O-arm. Indirect decompression is an excellent method, but shows some limitations. Indirect decompression reportedly fails to provide good results in severe central canal stenosis, gross motor deficit or cauda equina syndrome. With regard to fusion level, OLIF51 is more difficult than TLIF51 due to the vascular anatomy. This new technique also involves a slight learning curve problem for surgeons.

Several limitations to this study need to be kept in mind. First, the retrospective nature of the study necessarily involves the possibility of selection bias in procedures. Preoperative FH was higher in Group SO than in Group P/T, which may suggest selection bias for surgeons toward PLIF or TLIF for severely stenotic cases. Second, some of the weaker p-values might represent false-positive differences due to multiple testing. Third, Group P/T was smaller than Group SO, and increased sample sizes are needed to confirm our findings. Further prospective research is warranted to clarify the long-term clinical outcomes.

## 5. Conclusions

Simultaneous single-position O-arm-navigated OLIF-PPS reduces the surgical time, blood loss, and time to ambulation after operation without risking adverse events associated with intraoperative radiation exposure in operating staff. Good indirect decompression can be achieved with this method. Excellent clinical results of this technique were obtained without the direct decompression of MIS-PLIF/TLIF.

## Figures and Tables

**Figure 1 jcm-10-04938-f001:**
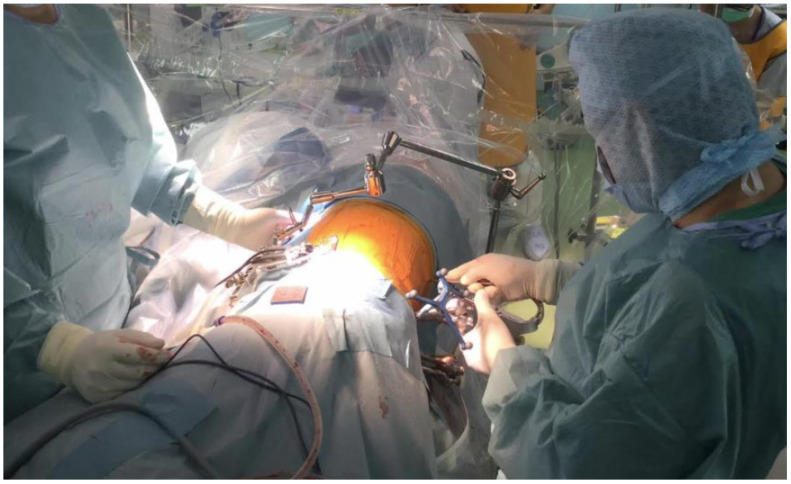
Intraoperative picture showing two surgeons operating simultaneously. The patient is in the right lateral position. The main surgeon is shown at the left, while on the right the assistant surgeon is applying a rod percutaneously.

**Figure 2 jcm-10-04938-f002:**
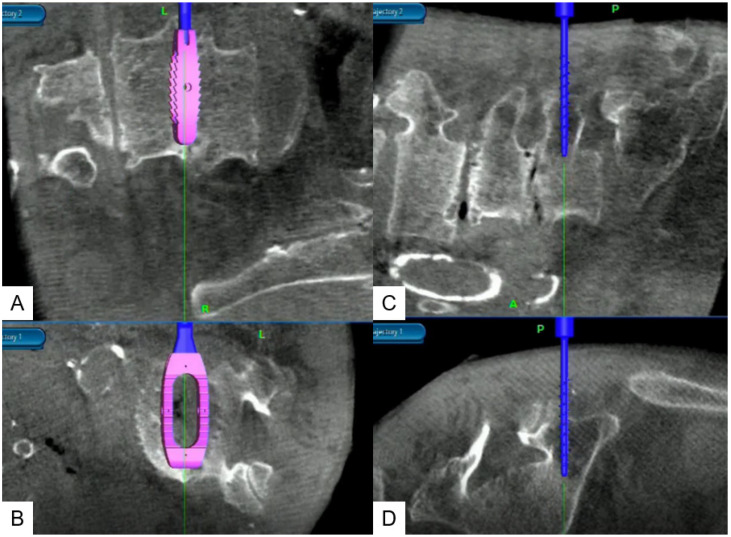
O-arm images showing navigated cage and screw placement. (**A**) Coronal view of the cage insertion; (**B**) Axial view of cage insertion; (**C**) Sagittal view of PS insertion; (**D**) Coronal view of PS insertion.

**Figure 3 jcm-10-04938-f003:**
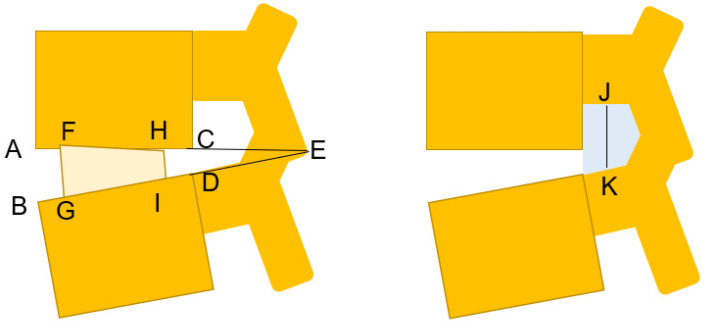
Radiological measurement. Disc height = (AB + CD)/2; segmental lordosis = angle AEB; cage height = (FG + HI)/2; CH—preoperative disc height = (FG + HI) − (AB + CD); foraminal height = JK; foraminal area = gray area. A: Antero-inferior corner; C: postero-inferior of the upper vertebra; B: antero-superior corner; D: postero-superior corner of the lower vertebra; J: center of the lower pedicle cortex; K: center of the upper pedicle cortex.

**Figure 4 jcm-10-04938-f004:**
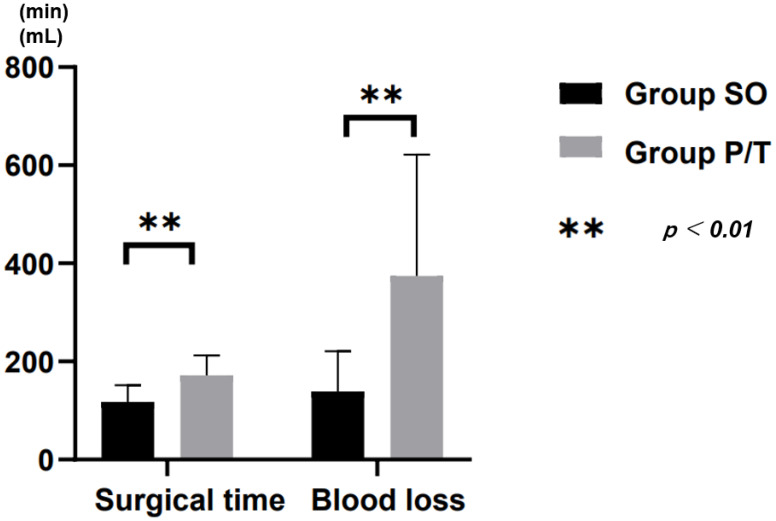
Surgical time and blood loss.

**Figure 5 jcm-10-04938-f005:**
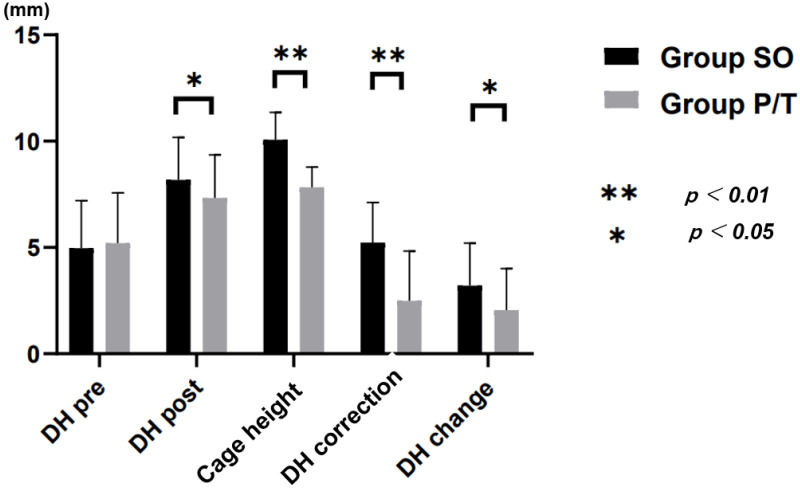
Radiological result. DH: disc height; CDR: cage disc ratio.

**Figure 6 jcm-10-04938-f006:**
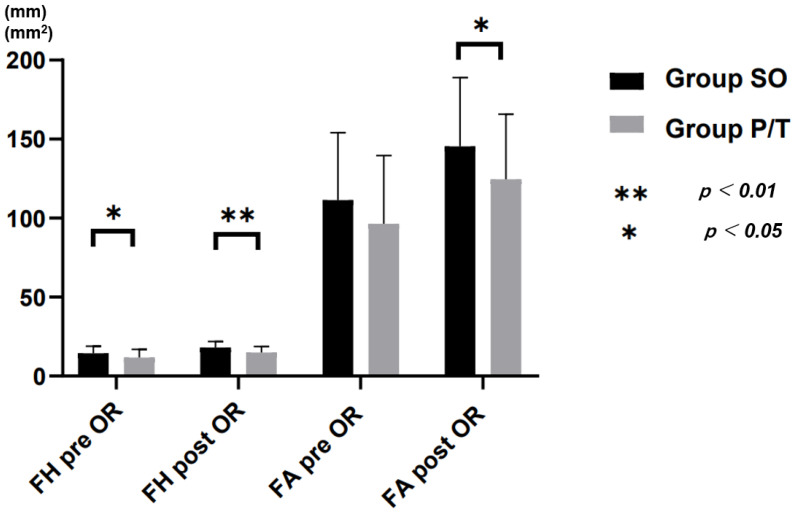
Radiological result. FH: foraminal height; FA: foraminal area; OR: operation.

**Figure 7 jcm-10-04938-f007:**
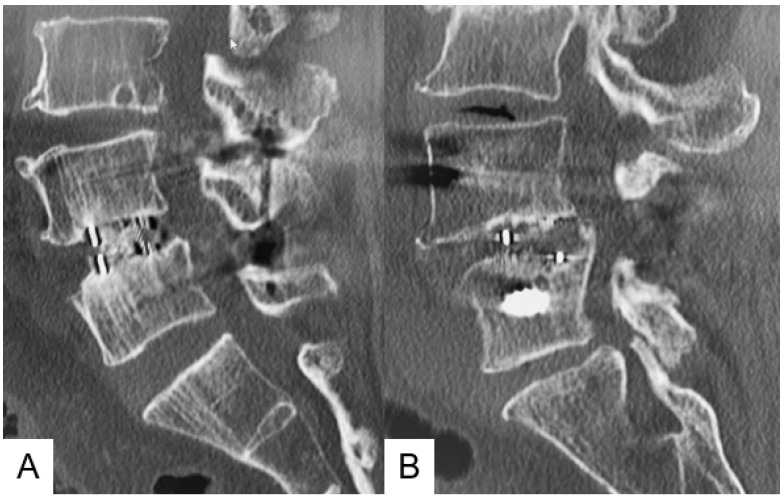
One-year follow-up CT images in both groups. (**A**): Group SO; (**B**): Group P/T.

**Table 1 jcm-10-04938-t001:** Demographic characteristics of patients.

	Group SO	Group P/T	*p*-Value
Number of patients	63	35	
Age (years)	68.6 ± 10.4	71.0 ± 13.5	0.802
**Gender**			
Male	20	15	
Female	43	20	
BMI (kg/m^2^)	23.6 ± 4.7	22.9 ± 2.9	0.552
**Disorder**			
Spondylolisthesis	49	28	
Lumbar stenosis	11	5	
Degenerative Disc Disease	3	2	
**Fused level of vertebra**			
L2/3	1	2	
L3/4	10	4	
L4/5	51	22	
L5/S1	1	7	

**Table 2 jcm-10-04938-t002:** Surgical and radiographic results for Group SO and Group P/T.

	Group SO	Group P/T	*p*-Value
Mean Follow up (month)	23.2 ± 6.9	27.9 ± 11.7	
Surgical time (min)	112.0 ± 32.4	171.8 ± 40.6	<0.001
Blood loss (ml)	139.2 ± 82.0	374.2± 247.7	<0.001
Mobilization time (day)	2.7 ± 1.0	3.9 ± 2.4	0.002
Cage height (mm)	10.1 ± 1.3	7.8 ± 1.0	<0.001
DH pre OR (mm)	5.0 ± 2.2	5.2 ± 2.4	0.992
FH pre OR (mm)	14.4 ± 4.5	11.7 ± 5.3	0.021
FA pre OR (mm^2^)	111.4 ± 42.6	96.4 ± 43.2	0.098
DH correction (mm)	5.2 ± 1.9	2.5 ± 2.3	<0.001
DH post OR (mm)	8.2 ± 2.0	7.3 ± 2.0	0.042
FH post OR (mm)	18.1 ± 3.7	14.9 ± 3.9	<0.001
FA post OR (mm^2^)	145.4 ± 43.5	124.6 ± 41.2	0.044
DH change (mm)	3.2 ± 2.0	2.1 ± 1.9	0.048
Segmental lordosis pre OR (degree)	10.3 ± 5.9	11.5 ± 8.4	0.404
Segmental lordosis post OR (degree)	10.0 ± 6.0	12.8 ± 8.1	0.201

DH: disc height; OR: operation; FH: foraminal height; FA: foraminal area; CDR: cage disc ratio.

**Table 3 jcm-10-04938-t003:** Clinical results for Group SO and Group P/T.

	Group SO	Group P/T	*p*-Value
Number of patients	63	35	
VAS of back pain	5.6 ± 2.4	6.0 ± 2.9	0.578
ODI	45.4 ± 17.7	55.4 ± 19.7	0.142
Complication	6.3%	14.1%	0.191
Neural injury	2	0	
Dural tear	0	2	
SSI	1	0	
Hematoma	0	2	
Reoperation	1	1	

ODI: Oswestry Disability Index; VAS: visual analogue scale; SSI: surgical site infection.

## Data Availability

The data presented in this study are available in the article.

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
