# Peer review of "Comparison of Simultaneous Single-Position Oblique Lumbar Interbody Fusion and Percutaneous Pedicle Screw Fixation with Posterior Lumbar Interbody Fusion Using O-arm Navigated Technique for Lumbar Degenerative Diseases"

_jcm, 2021, doi:10.3390/jcm10214938_

Round 1

Reviewer 1 Report

Dear authors,

Thank you for your very interesting manuscript, it is a very nice demonstration of a novel surgical strategy. The research questions are basic but right, the chosen research design is unfortunately retrospective but probably the best data currently available. The study is well carried out. My comments are mainly on improving readability and textual presentation. Good luck!

- line 17-19: the abstract should not be divided in separate sections with numbers like (1) and (2), and also no have subtitles like background/methods. Furthermore the background directly starts with your ‘study aim’. I would suggests having 1 or 2 sentences to outline the current standards in surgery for the lumbar degenerative spine and introduce the novelty of your method.

- line 19: ‘minimally invasive’ should be written in full before ‘MI’.

- line 19: the first line ‘this is retrospective comparative …’ reads poorly.

- line 21: it is of course a retrospective study, therefore please specify if 1 year follow-up was an ex or inclusion criteria, OR if the percentage of 1 year follow-ups was an outcome/result. If this is the case, the mean/median follow-up should be given in the results.

- line 22: it should better be: ‘in the SO group compared to the P/T group, the surgical time was 118 versus 172 minutes (p<0.01).’ I would furthermore suggest not having so many decimals, for both the mean and the p-value, Throughout the whole abstract and rest of the paper.

- line 26: suggestion for re-writing the sentence: ‘In the SO-group the fusion rate was 96.8%, similar compared to 94.2% in the P/T-group (p=0.985).’

- line 27: suggestion for re-writing the sentence: ‘The complication rate was 6.3% in the SO group compared to 14.1% in the P/T-group (p=0.191)’. The P-value from the chi-square should be there.

- Line 28: the sentence part ‘.. O-arm navigated C-arm free..’ is confusing, also in the title, I would suggest to leave C-arm out and/or to rephrase this.

- Line 29: the radiation concerns part is perhaps something for the discussion only and not for the abstract.

- Line 36: should read: ‘.. may require fusion surgery..’.

- Line 39: are there probably more severe complications than just back pain?

- Line 44: should read:  ‘…morbidity and less postoperative pain …’

- Line 47: I think you should stress that the patients body positioning on the operating table is required to change. Which is indeed suboptimal.

- Line 51: are there very few studies or are there none studies? If there are, you should refer to it or otherwise state there are none.

- Line 53: a suggestion for better reading: ‘The aim of this study is exactly that, to retrospectively compare clinical, surgical and radiographic outcomes for both techniques’

- Line 56: was there written informed consent gathered for all patients or was that waived by the REB since it was a retrospective study? Please specify and make sure it is in line with the ‘statements’ section at the end of the manuscript.

- Line 56-61: Were there any exclusion criteria? How many were considered but excluded? As mentioned about the abstract, please specify if 1 year follow-up was an ex or inclusion criteria, OR if the percentage of 1 year follow-ups was an outcome/result. If this is the case, the mean/median follow-up should be given in the results. The patient inclusion/exclusion should be much more clear and concise.

- Line 63-82: This is a nice and clear explanation of the procedure. I would suggest to use ‘articles’ like ‘the’ in the needed places in the sentences.

- Line 81: a suggestion: ‘The complete procedure is described in a technical note, published in 2019.[9]’

- Line 86-87: ’surgeon’ instead of ‘surgen’ and ‘applying‘ instead of ‘appling’

- Line 101-103: perhaps add the scaling options for the VAS and ODI, and perhaps a reference for ODI since this might not be familiar to all surgeons.

- Line 101-108: this is a retrospective study, so are sections 2.3 and 2.4 retrieved from the patients electronic health record? Are all parameters standardly noted for all patients? Was was done when there was missing data? How where the records scanned? Was there more than one observer for the patients records as was for the radiographic measurements, I think this whole section deserves some specification to improve reproducibility of your data.

- Line 110-121: Great explanation and figure 3. Furthermore it is in cursive text, please correct.

- Line 129: write ‘SD’ in full as ‘standard deviation’

- Line 130: why did you test all the continuous variables with a Mann-Whitney-U? Did you test for normality of distribution? Was there any reason not to use a t-test? Furthermore, you performed ~20 comparative tests, shouldn’t you correct for multiple testing? Than p-values like 0.042 and 0.048 for DH are probably not significant.

- Table 2: great overview, perhaps make a subheading for clinical/surgical/radiographic. Also I would suggest to use 3 decimals max and P<0.001 as the lowest optional value.

- Line 134 – 184: I would suggest subheadings for the text just as you did for the methods section.

- Figure 7: please refer to it somewhere in the manuscript, and please specify the aim of this figure, what is important to look at?

- Line 194: please add ‘and’ just before ‘… less incidence of dural tears’, and also references to literature.

- Line 195: ‘profile’ instead of ‘profil’

- Line 186-246: please check the discussion and conclusion on the missing of ‘articles’ like ‘the’ and ‘a’ and add these, for improved readability.

- Line 227-230: is there any literature that quantified radiation doses for both O-arm and C-arm and support the claim for the superiority of one over the other?

- Line 232-240: great that you’ve addressed your limitations, to add two more scientific limitations: the first and most important is to stress the retrospective nature of the study, explain the possibility of different types of bias and your suggestion for prospective trials. The second is to stress that you did a lot of statistical comparing tests and no correction for multiple testing, therefore many of the weak p-values might false positive differences.

- Line 242-246: as mentioned for in the title and abstract, is it necessary to stress: ‘..C-arm free..’ ? Because it complicates the already long and complicated title even more.

Author Response

We appreciate your valuable comments and efforts.

To reviewer 1

>- line 17-19: the abstract should not be divided in separate sections with numbers like (1) and (2), and also no have subtitles like background/methods. Furthermore the background directly starts with your ‘study aim’. I would suggests having 1 or 2 sentences to outline the current standards in surgery for the lumbar degenerative spine and introduce the novelty of your method.

We appreciate your wonderful suggestion. We changed abstract as follows.

Abstract: Minimally invasive posterior or transforaminal lumbar interbody fusion (MI-PLIF/TLIF) are widely accepted procedures for lumbar instability due to degenerative or traumatic diseases. Nowadays, oblique lateral interbody fusion (OLIF) is receiving considerable attention because of its less damage to back muscle and neural tissue. The aim of this study is to compare the clinical and radiographic outcomes of simultaneous single-position OLIF and percutaneous pedicle screw (PPS) fixation with MI-PLIF/TLIF. This is retrospective comparative study of 98 patients, 63 in single position OLIF (group SO) and 35 patients in MI-PLIF/TLIF (group P/T). More than 1 year follow up cases were included in this study. Mean follow up period for group SO was 32.9 ± 7.0 months and that for group P/T was 33.7 ± 7.5 months Clinical and radiological evaluation were performed. In the SO group compared to the P/T group, the surgical time and blood loss were 118 versus 172 minutes (p<0.01) and 139 versus 374 ml (p<0.01), respectively. The cage height, the change of disk height, and the postoperative foraminal height were high with statistical significance in group SO compared to group P/T. The fusion rate of group SO and group P/T at one year follow-up were 96.8% and 94.2%, respectively (p =0.985). Total complication rate in group SO was 6.3% while that in group P/T was 14.1%. Simultaneous single position O-arm navigated OLIF reduces the surgical time, blood loss, ambulation time after operation without the risk of an adverse event of intraoperative radiation to operating staff. Good indirect decompression can be achieved with this method

>- line 19: ‘minimally invasive’ should be written in full before ‘MI’.
We changed as you mentioned.

>- line 19: the first line ‘this is retrospective comparative …’ reads poorly.

We changed as you mentioned.

>- line 21: it is of course a retrospective study, therefore please specify if 1 year follow-up was an ex or inclusion criteria, OR if the percentage of 1 year follow-ups was an outcome/result. If this is the case, the mean/median follow-up should be given in the results.

We changed as you mentioned.

>- line 22: it should better be: ‘in the SO group compared to the P/T group, the surgical time was 118 versus 172 minutes (p<0.01).’ I would furthermore suggest not having so many decimals, for both the mean and the p-value, Throughout the whole abstract and rest of the paper.

We changed as you mentioned.

>- line 26: suggestion for re-writing the sentence: ‘In the SO-group the fusion rate was 96.8%, similar compared to 94.2% in the P/T-group (p=0.985).’

We changed as you mentioned.

>- line 27: suggestion for re-writing the sentence: ‘The complication rate was 6.3% in the SO group compared to 14.1% in the P/T-group (p=0.191)’. The P-value from the chi-square should be there.

We changed as you mentioned.

>- Line 28: the sentence part ‘.. O-arm navigated C-arm free..’ is confusing, also in the title, I would suggest to leave C-arm out and/or to rephrase this.

Thank you for your advice. We changed as you mentioned.

>- Line 29: the radiation concerns part is perhaps something for the discussion only and not for the abstract.

We changed as you mentioned.

>- Line 36: should read: ‘.. may require fusion surgery..’.

We changed as you mentioned.

>- Line 39: are there probably more severe complications than just back pain?

We changed the sentence as you mentioned.

which in long term may lead to chronic back pain and neurological defect

>- Line 44: should read:  ‘…morbidity and less postoperative pain …’

We changed the sentence as you mentioned.

>- Line 47: I think you should stress that the patients body positioning on the operating table is required to change. Which is indeed suboptimal.

We changed the sentence as you mentioned.

The patient body repositioning on the operating table is required, which is indeed suboptimal and increases operative time and medical cost [7].

>- Line 51: are there very few studies or are there none studies? If there are, you should refer to it or otherwise state there are none.

We changed the sentence as you mentioned.

There are no studies

>- Line 53: a suggestion for better reading: ‘The aim of this study is exactly that, to retrospectively compare clinical, surgical and radiographic outcomes for both techniques’

We changed the sentence as you mentioned.

>- Line 56: was there written informed consent gathered for all patients or was that waived by the REB since it was a retrospective study? Please specify and make sure it is in line with the ‘statements’ section at the end of the manuscript.

Thank you for your advice. We wrote that in the statement section.

>- Line 56-61: Were there any exclusion criteria? How many were considered but excluded? As mentioned about the abstract, please specify if 1 year follow-up was an ex or inclusion criteria, OR if the percentage of 1 year follow-ups was an outcome/result. If this is the case, the mean/median follow-up should be given in the results. The patient inclusion/exclusion should be much more clear and concise.

Thank you for your advice. We changed the sentence as follows.

We retrospectively evaluated lumbar interbody fusion patients from May 2017 to January 2020. The inclusion criteria were one level fusion, more than one year follow up. The exclusion criteria were infection and spinal tumor. There were 98 patients with lumbar degenerative disease were matched with those criteria;

>- Line 63-82: This is a nice and clear explanation of the procedure. I would suggest to use ‘articles’ like ‘the’ in the needed places in the sentences.

Thank you for your advice. We changed the sentence as you mentioned.

>- Line 81: a suggestion: ‘The complete procedure is described in a technical note, published in 2019.[9]’

Thank you for your advice. We changed the sentence as you mentioned.

>- Line 86-87: ’surgeon’ instead of ‘surgen’ and ‘applying‘ instead of ‘appling’

Thank you for your advice. We changed the sentence as you mentioned.

>- Line 101-103: perhaps add the scaling options for the VAS and ODI, and perhaps a reference for ODI since this might not be familiar to all surgeons.

Thank you for your advice. We added the reference [10].

>- Line 101-108: this is a retrospective study, so are sections 2.3 and 2.4 retrieved from the patients electronic health record? Are all parameters standardly noted for all patients? Was was done when there was missing data? How where the records scanned? Was there more than one observer for the patients records as was for the radiographic measurements, I think this whole section deserves some specification to improve reproducibility of your data.

Thank you for your comment. As you mentioned, all patients records are in the electric medical record. All data including surgical time, blood loss, complications were recorded in electric medical record.

The senior doctors measured twice and get average data.

>- Line 110-121: Great explanation and figure 3. Furthermore it is in cursive text, please correct.

Thank you for your advice. We changed the sentence as you mentioned.

>- Line 129: write ‘SD’ in full as ‘standard deviation’
Thank you for your advice. We changed the sentence as you mentioned.

>- Line 130: why did you test all the continuous variables with a Mann-Whitney-U? Did you test for normality of distribution? Was there any reason not to use a t-test? Furthermore, you performed ~20 comparative tests, shouldn’t you correct for multiple testing? Than p-values like 0.042 and 0.048 for DH are probably not significant.

Thank you for your comment. Our data was non-normal distribution. So we had to go for Mann-Whitney-U test instead of independent t test.

You are absolutely right. P-values of 0.0025 should be significant. Even such situation, surgical time, blood loss, cage height, and CDR were significantly different.

> Table 2: great overview, perhaps make a subheading for clinical/surgical/radiographic. Also I would suggest to use 3 decimals max and P<0.001 as the lowest optional value.

Thank you for your advice. We changed the table 3 as you suggested.

> Line 134 – 184: I would suggest subheadings for the text just as you did for the methods section.

Thank you for your advice. We changed the sentence as you mentioned.

> Figure 7: please refer to it somewhere in the manuscript, and please specify the aim of this figure, what is important to look at?

Thank you for your advice. We added the sentence as you mentioned.

> Line 194: please add ‘and’ just before ‘… less incidence of dural tears’, and also references to literature.

Thank you for your advice. We changed the sentence and added the reference

> Line 195: ‘profile’ instead of ‘profil’

We corrected the word.

> Line 186-246: please check the discussion and conclusion on the missing of ‘articles’ like ‘the’ and ‘a’ and add these, for improved readability.

We corrected the sentences.

> Line 227-230: is there any literature that quantified radiation doses for both O-arm and C-arm and support the claim for the superiority of one over the other?

Thank you for your advice. We added the sentence as follows.

One screw placement under fluoroscopy takes 7-20 seconds [Z]. Thus, overall exposure to patient is equivalent.

  1. Jones DPG, Robertson PA, Lunt B, Phys M, Jackson SA. Radiation Exposure During Fluoroscopically Assisted Pedicle Screw Insertion in the Lumbar Spine. Spine. 2000;25:1538-1541.

> Line 232-240: great that you’ve addressed your limitations, to add two more scientific limitations: the first and most important is to stress the retrospective nature of the study, explain the possibility of different types of bias and your suggestion for prospective trials. The second is to stress that you did a lot of statistical comparing tests and no correction for multiple testing, therefore many of the weak p-values might false positive differences.

Thank you for your advice. We changed the sentence as follows.

There are several limitations of our study. First, this is the retrospective nature of the study, there are some possibilities of procedure selection bias. The preoperative FH was higher in group SO than group P/T, this can be selection bias of surgeon towards PLIF or TLIF for severely stenotic cases. Second, some of the weak p-values might false positive differences due to multiple testing. Third, the sample size in group P/T was relatively smaller than in group SO, with increased sample size required to confirm our findings. Further prospective research is warranted to clarify the long-term clinical outcome.

> Line 242-246: as mentioned for in the title and abstract, is it necessary to stress: ‘..C-arm free..’ ? Because it complicates the already long and complicated title even more.

Thank you for your comment. We deleted that word.

Reviewer 2 Report

It is my pleasure to give my insights and highlight some recommendations and commentaries to the manuscript, considering my expertise in the field. I really appreciate this opportunity to discuss the evidence and I hope my contributions could serve to improve the final version of the study.

The manuscript titled “Comparison of simultaneous single-position oblique lumbar interbody fusion and percutaneous pedicle screw fixation with posterior lumbar interbody fusion using O-arm navigated C-arm free technique for lumbar degenerative diseases.” deals with an important issue of the surgery methods to lumbar degenerative disease. The aim of this study was to compare the clinical and radiographic outcomes of simultaneous single-position oblique lateral interbody fusion and percutaneous pedicle screwfixation with posterior/transforaminal lumbar interbody fusion.

This paper is very interesting in this field of research. It is well conducted and suitable with the remit and purpose of the journal standard. 

I believe that although the authors have carried out an interesting study and the content of the document has high potential and may be of interest to the field of study, there are some details that could be improved in order to increase its relevance, clarity, application practical and overall quality. 

Please revise the English language, some sentences are incorrect. 

Page 2, line 53, Please clearly and correctly clarify the purpose of the study.

2.3, line 103, This section repeats with the next section in “intraoperative or postoperative complications”.

2.5, line 111, please clarify “the cage to disc ratio” or “the disc to cage ratio”.

Table 2, please add the percentile increase of DH, FH and FA as described in line 157.

Page 6, line 154, please correct “p < 0.021” and “was significantly more”.

Figure 5, please adjust the location of “CDR”.

Figure 6, please add the percentile increase of DH, FH and FA as described in line 157.

Page 8, line 194, please add adequate reference.

Page 8, line 199, line 220, please clearly and correctly clarify the language.

Page 9, line 245, Please provide a clear “take-home message” of the importance of this paper. Be careful not to use words such as “group” related to the design of this study.

Author Response

We appreciate your efforts and comments.

To reviewer 2

> Page 2, line 53, Please clearly and correctly clarify the purpose of the study.

Thank you for your important comment. We changed the sentence as bellows.

The aim of this study is exactly that, to retrospectively compare clinical, surgical and radiographic outcomes for both techniques.

>2.3, line 103, This section repeats with the next section in “intraoperative or postoperative complications”.

We appreciate your comment. We deleted that sentence.

>2.5, line 111, please clarify “the cage to disc ratio” or “the disc to cage ratio”.

We are sorry. It should be cage height-preoperative disc height (CH-DH pre), which means relatively how large cage could we put into the collapsed disc space (how much sagittal correction could we get). We corrected the sentences as follows.

The disc height correction was calculated by CH-DH preop.

>Table 2, please add the percentile increase of DH, FH and FA as described in line 157.

We added the percentile increase of DH, FH and FA.

>Page 6, line 154, please correct “p < 0.021” and “was significantly more”.

>Figure 5, please adjust the location of “CDR”.

We corrected as you mentioned.

>Figure 6, please add the percentile increase of DH, FH and FA as described in line 157.

Thank you for your valuable comment. However, we added these in Table 3 and we already mentioned in the manuscript. Could you excuse us not to put the percentile of those? Because it may become very busy figure.

>Page 8, line 194, please add adequate reference.

We added the reference as you mentioned.

>Page 8, line 199, line 220, please clearly and correctly clarify the language.

According to your advice, we corrected the sentence as bellows.

OLIF has received considerable attention as it provides clear operative field, reduces neurological complications, and the navigated implants are available.

>Page 9, line 245, Please provide a clear “take-home message” of the importance of this paper. Be careful not to use words such as “group” related to the design of this study.

We appreciate your important comment. We changed the sentence as bellows.

Excellent clinical results of this technique were obtained without direct decompression of MIS-PLIF/TLIF.

Reviewer 3 Report

The authors compare OLFIF with PLIF and use an O-arm navigated technique for the positioning of the implants.

The study has a retrospective design and is thoroughly performed, the manuscript is straight forward written. The authors conclude that the O-arm use for implantation of OLIF reduces the surgical time, blood loss and inpatient stay.

Clinical results of OLIF vs. PLIF were alike, this confirms other already existing literature. O-arm navigation reduces radiation exposure to almost zero for the operating staff. However, the authors don´t mention learning curve effects of the navigated surgical technique. Maybe they can state for how long the conduct lumbar surgeries with O-arm vs. conventional C-arm.

Please correct:

l. 23 blood loss were 177 ml (NOT min).

Figure 4&5 "p" for statistical significance should not be capitalized.

ll. 242 simultaneous "S" should not be capitalized

Did the authors see differences in the various levels of vertebral fusion? Especially, L5/S1 was only one SO performed. This is a more difficult level to operate than the upper lumbar spine. Was this the reason to choose the PLIF in 7 cases, were as in total it were almost 50% less PILF than OLIF. – Please discuss this aspect.

Author Response

We appreciate your efforts and comments.

To reviewer 3

>Clinical results of OLIF vs. PLIF were alike, this confirms other already existing literature. O-arm navigation reduces radiation exposure to almost zero for the operating staff. However, the authors don´t mention learning curve effects of the navigated surgical technique. Maybe they can state for how long the conduct lumbar surgeries with O-arm vs. conventional C-arm.

Thank you for your important comment.

We perform this new simultaneous O-arm technique from 2018. For MIS-PLIF/TLIF technique under C-arm, we have more than 10-year-experience.

According to your advice, we added the sentence in discussion section as bellows.

For this new technique, there is a slight learning curve problem for the surgeons as well.

>l. 23 blood loss were 177 ml (NOT min).

>Figure 4&5 "p" for statistical significance should not be capitalized.

>ll. 242 simultaneous "S" should not be capitalized

We corrected those as you mentioned.

>Did the authors see differences in the various levels of vertebral fusion? Especially, L5/S1 was only one SO performed. This is a more difficult level to operate than the upper lumbar spine. Was this the reason to choose the PLIF in 7 cases, were as in total it were almost 50% less PILF than OLIF. – Please discuss this aspect.

We appreciate your important comment. As you mentioned, OLIF51 is a relatively difficult compared with TLIF51. We added the sentences in discussion section as bellows.

With regard to fusion level, OLIF51 is more difficult than TLIF51 due to the vascular anatomy. This new technique also involves a slight learning curve problem for surgeons.
